# Yeast-Derived β-Glucan in Cancer: Novel Uses of a Traditional Therapeutic

**DOI:** 10.3390/ijms20153618

**Published:** 2019-07-24

**Authors:** Anne Geller, Rejeena Shrestha, Jun Yan

**Affiliations:** 1Department of Microbiology and Immunology, University of Louisville School of Medicine, Louisville, KY 40202, USA; 2Immuno-Oncology Program, Division of Immunotherapy, Department of Surgery, The James Graham Brown Cancer Center, University of Louisville School of Medicine, Louisville, KY 40202, USA

**Keywords:** yeast-derived β-Glucan, cancer, immunotherapy, combination therapy, trained immunity, metabolic reprogramming, adjuvant

## Abstract

An increased understanding of the complex mechanisms at play within the tumor microenvironment (TME) has emphasized the need for the development of strategies that target immune cells within the TME. Therapeutics that render the TME immune-reactive have a vast potential for establishing effective cancer interventions. One such intervention is β-glucan, a natural compound with immune-stimulatory and immunomodulatory potential that has long been considered an important anti-cancer therapeutic. β-glucan has the ability to modulate the TME both by bridging the innate and adaptive arms of the immune system and by modulating the phenotype of immune-suppressive cells to be immune-stimulatory. New roles for β-glucan in cancer therapy are also emerging through an evolving understanding that β-glucan is involved in a concept called trained immunity, where innate cells take on memory phenotypes. Additionally, the hollow structure of particulate β-glucan has recently been harnessed to utilize particulate β-glucan as a delivery vesicle. These new concepts, along with the emerging success of combinatorial approaches to cancer treatment involving β-glucan, suggest that β-glucan may play an essential role in future strategies to prevent and inhibit tumor growth. This review emphasizes the various characteristics of β-glucan, with an emphasis on fungal β-glucan, and highlights novel approaches of β-glucan in cancer therapy.

## 1. Introduction

Though fungi did not emerge as a major pathogen for mammals until the late 20th century, animals have been fighting the harmful effects of fungi for millennia, and, thus, over the course of evolution have evolved robust innate immune responses to components of the fungal cell wall, such as β-glucan [1,2]. Naturally, occurring polysaccharides, such as β-glucan, have been widely studied in health and human disease in both western and eastern medicine. To date, almost 15,000 scientific articles mention β-glucan. Β-glucans are biologically active polymers of glucose found in the cell wall of bacteria (such as the *Agrobacterium* species), fungi (such as *Aspergillis*), algae, edible mushrooms, cereal grains, oats, barley, wheat and rye [3,4]. Yeast-derived β-glucan has a linear backbone of β(1,3)-linked D-glucose molecules with β(1-6) side chains of variable lengths occurring at different locations along the backbone of the polymer. There are several known configurations of β-glucan linkages, including β(1,3), β(1,4) and β(1.6), however only molecules with a β(1,3)-linked D-glucose backbone has shown immunomodulatory and stimulatory activity, which is why it has been classified as a biological response modifier (BRM) [5] 

Though various glucose polymers have been studied for some time in traditional medicine, it was not until around 1941 when β-glucan began to be identified as a potent immune modulator in western medicine. In a 1941 paper by Pillemer and Ecker, they described zymosan, a crude yeast cell wall preparation, that acted as a modulator of non-specific immune responses, which we now know as the complement system [6,7]. Though at the time the active component of zymosan was unknown, it was later discovered that the active biological properties of zymosan were due mainly to β-glucan [8]. Since then, β-glucan has become an extremely popular topic of research both in the worlds of infectious diseases and tumor immunology. With the discovery of specific receptors that directly mediate the stimulatory activity of β-glucan on immune cells, such as Dectin-1 and Complement Receptor 3 (CR3), along with more recent research on how β-glucan mediates an innate memory response called “trained immunity,” and the use of β-glucan in combination with immunotherapy, there has been a reinvigorated wave of interest in β-glucan and anti-tumor responses [6].

## 2. The Structure and Signaling of β-Glucan

β-glucans are among the most abundant type of polysaccharide within the cell wall of fungus and bacteria. Though all β-glucans consist of glucose molecules linked together by a (1-3), (1,4) or (1,6) linear β-glycosidic chain, the types of glucans vary in length and branching structures, where branches may attach to the backbone at different positions and consist of one to many monosaccharaide units [9]. By definition, the D-glucopyranose polymers that constitute a β -glucan are linked in a β anomeric configuration, while α-D-glucans would have their D-glucopyranose units connected in an α configuration. Branching from the core glycosidic chains are usually (1-4) or (1-6), where fungal β-glucans usually have a (1-3) linked backbone with (1-6) side branches, while bacteria tend to have (1-4) side branches [10]. Though most glucans used in medical research have a (1-3) backbone with (1-6) linkages, fungal β-glucans themselves have a rich structural diversity and have been isolated from a number of fungal sources. For example there have been fungal glucans that have been identified to possess a backbone of linked (1-6)-β units with (1-3)- β branches [11,12]. There also exists a simple linear (1-3)- β-D-glucan which can be isolated from *Poria cocos* and *Saccharomyces cerevisiae*, and a linear water-soluble (1-6)- β-D-glucan which can be isolated from *Lasallia pustulata*. A linear mixed-linkage (1-3),(1-4)- β-D-glucan has been identified that is closely related to the structure of cereal β-D-glucans [9], and fungal β-D-glucan bearing a (1-3) backbone and (1-4) branching structures have also been isolated [13]. To add to this complexity, glucans have been discovered that have alternating α- and β-glycosidic linkages, and those that have a (1,6) β—D-glucan backbone, but α (1,4) branches at intervals of three units along the backbone [13,14]. Ultimately, there are a wide range of β-D-glucans that can be isolated from fungal sources, and characterizing the precise composition of a β-D-glucans is of primary importance in glucan research.

Research using NMR is capable of determining the exact length of side branches, which gives great insight into the interactions of β-glucan. For example, Lowman et al. showed that in *Candida* glabrate, the (1-6) β-linked side chins have an average length of 4−5 repeat units spaced every 21 repeat unites along with the (1-3)-linked core [15]. It has been shown that the side chains are vital for the conformation of β-glucan in aqueous solutions, where β-glucans will form a triple helix, single helix, or coils [16]. It has also been suggested that the specific conformation of β-glucan is essential for its ability to act as an immunopotentiator and that a higher degree of complexity is associated with enhanced anti-cancer properties [10]. For this reason, using NMR studies to better characterize the differences between the glucans could lead to a better understanding of how structure relates to functionality, and also why different β-glucans behave differently.

Aside from the differential branching and glyosidic properties of β-glucan, several other properties also differentiate them from one another. The solubility, purity, molecular mass, tertiary structure, degree of branching, charge of the polymer, and solution conformation all have a strong influence on the immune-modulatory effect of the particle [17].

Fungal β-glucans function as a Pathogen Associated Molecular Pattern (PAMP), where host immune responses are triggered following contact. There have been several membrane-bound receptors involved in β-glucan signaling as a PAMP, which include scavenger receptors [18], lactosylceramide [19], CR3; CD11b/CD18 [20], and dectin-1 [21,22]. Dectin-1 is the most commonly studied β-glucan receptor and is a type II transmembrane protein C-type lectin receptor found on monocytes, macrophages, neutrophils and dendritic cells (DCs). When Dectin-1 is bound by β-glucan, it induces phosphorylation of its intracellular immunoreceptor tyrosine-based activation motif (ITAM) [23] and Syk in addition to activation of a PI3K/Akt pathway [24]. This ultimately causes phagocytosis, ROS generation, microbial killing and cytokine production [25,26]. CR3, also known as Macrophage 1 Antigen (Mac-1) is a heterodimeric transmembrane glycoprotein comprised of CD11b noncovalently associated with CD18 and belongs to the family of β_2_ integrins [27]. It is found on neutrophils, monocytes and natural killer (NK) cells. CR3 is unique in that it has two distinct binding sites for ligands. The first is an I-domain that binds ligands, such as extracellular matrix proteins, iC3b, and intracellular adhesion molecules like intracellular adhesion molecule 1 (ICAM-1). The second ligand binding site is a carbohydrate-binding lectin-like domain that is capable of binding β-glucan [28]. Β-glucan binding to CR3 will result in enriched cytotoxicity against iC3b-opsonized target cells such a tumor cells, phagocytosis and degranulation [29].

Despite the expanding research regarding β-glucan, a significant challenge of β-glucan research is the high structural variability, low purity, and promiscuity of receptor activation. The ability of β-glucan to activate multiple receptors results in multiple and variable signaling pathways. Additionally, differences in molecular weight, degree of branching, polymeric charge and solution conformation affect the degree of triple helical formation, which affects the solubility of β-glucan. Together these variables can cause β-glucans isolated from the same source to take on drastically different immune-modulating properties [17,27,30]. With such variation at hand, incorrect conclusions can be drawn when comparing β-glucans from different sources or with disparities in purity. Even further, many scientists use extraction techniques involving the use of high or low pH and temperature conditions which can lead to denatured or renatured molecules that may have a different conformation and thus different immune-stimulatory properties [31,32]. For this reason, a great deal of care must be taken to not only isolate and purify β-glucan, but also to confirm the purity status of these extracts before use and throughout experimenation. It also becomes important to clearly and meticulously define and report the source of β-glucans used in scientific research and to catalog the preparation and treatment of all samples used. 

## 3. The Trafficking of β-Glucan

The overall efficacy and immune-modulatory properties of β-glucan ultimately depends on adequate delivery of β-glucan throughout the body and to the immune cells at distant lymphoid organs. For this reason, understanding the trafficking mechanisms of β-glucan has been a major focus of β-glucan related research. Of all of the routes of administration, orally administered β-glucan has been the most extensively studied, however intravenous (IV) and intraperitoneal (IP) injections of β-glucan have also been used. Following oral administration, β-glucans move into the proximal small intestine where they are phagocytosed by intestinal epithelial cells or pinocytic microfold cells (M-cells) which transport β-glucan from the intestinal lumen to immune cells within Peyer’s patches [17,33,34]. Following exposure to β-glucan, gastrointestinal macrophages then migrate through the bloodstream toward the lymph system, ultimately resulting in trafficking to the bone marrow, spleen and lymph nodes [2,35]. In the lymph nodes, orally administered β-glucan activate DCs which capture dying tumor cells in vivo in the tumor setting and activate antigen-specific CD4 and CD8+ T-cells [36]. In the bone marrow, degradation products of β-glucan cause activation of neutrophils through the binding to CR3 and the modulation of hematopoietic myeloid progenitors within the bone marrow [37]. These can then induce CD3 dependent cellular cytotoxicity (CR3-DCC) in the presence of opsonized iC3b tumors [38].

As a side note, due to these changes in myelopoiesis, mice were shown to have protection from chemotherapy-induced myelosuppression, which further makes a case for combination therapy of β-glucan with standard chemotherapy regimens [39]. Overall, oral administration of β-glucan has been shown to result in increased phagocytic activity and oxidative bursts, along with increased IL-1, IL-6 and TNFα production in murine peritoneal macrophages [40]. Further, the acute phase of the humoral immune responses is also impacted, where β-glucan feeding resulted in increased lysozyme and ceruloplasmin activity in rats [41]. Human studies may show less promising results however, where following oral β-glucan consumption for seven days, researchers did not observe changes in cytokine production or microbicidal activity of leukocytes, and also were not able to detect β-glucan itself in the serum of volunteers [42].

Orally and IV-administered β-glucan are known to function similarly to enhance the tumoricidal activity of the immune response where following IV soluble β-glucan administration, β-glucan is delivered directly to the bone marrow and tissue macrophages. Here intact β-glucan is first taken up by macrophages and then cleaved into a 25-kDa active fragment. This active fragment binds to neutrophil CR3, which then primes effector cells for the targeted killing of tumor cells through the activation of a CR3-Syk-Phosphatidylinositol-3-Kinase (PI3K) signaling pathway. Ultimately, this indicates that the breakdown of β-glucan into active fragments is necessary for immune modulation [43,44]. Though oral administration may be considered safer and less invasive, several studies have shown IV-administered β-glucan is safe, well-tolerated by patients, and capable of enhancing immune responses in various settings [45,46,47]. Li et al. showed that one day following IV-administration of β-glucan, it could be found in splenic macrophages, but not neutrophils. After seven days, however, β-glucan-positive macrophages were absent from the spleen, while about 10% of neutrophils in both the spleen and the bone marrow were β-glucan positive [43].

Though IP injection of β-glucan has been less extensively studied than oral administration, there are some studies, which show that IP injection has more profound effects on immunological activity than does oral administration [48]. According to one study, using a triple helical β-1,6-branched β-1,3-glucan purified from *Lentinus edodes*, or a shitake mushroom, it was found that IP injection of the β-glucan resulted in activation of peritoneal macrophages showing increased TNFα production, and proliferation of peritoneal macrophages, whole spleen cells and lymphocytes. Zheng et al. also found that following IP administration, β-glucan was ingested by resident peritoneal macrophages which then trafficked to the lymph nodes, thymus and the tumor site in BALB/c mice bearing subcutaneous H22 hepatocellular carcinomas [49,50].

## 4. β-Glucan in Innate and Adaptive Immune Systems

The tumor microenvironment (TME) is a complex intercellular network of tumor cells and non-transformed host components (immune cells, endothelial cells, fibroblasts) driven by a dynamic network of soluble factors like cytokines, chemokines, growth factors and extracellular matrix proteins [51,52]. Host immune cells contribute a major proportion of the non-transformed component of the TME and are actively recruited by the tumor cells to alter the immune cell phenotype/function and promote either immune suppression or tolerance [53]. Integration of immune cells and other host-derived molecules to maintain a suppressive TME plays an important role in tumor growth and metastasis [54]. Modulation of the TME from an immune-suppressive to an inflammatory environment thus has a very promising potential as an anti-cancer therapy [55].

Fungal β-glucans have the potential to effectively manipulate the TME by influencing both innate and adaptive immune responses [56]. Once innate cells are activated through β-glucan binding to PRRs, as described above, they are recruited to further educate and activate adaptive immune cells. Fungal β-glucans thus have the ability to bridge the innate and adaptive immune responses [36,57,58].

As described briefly above, low molecular weight yeast-derived soluble β-glucans have been reported to bind with high affinity to CR3 expressed by innate cells, such as macrophages, dendritic cells, natural killer cells and neutrophils. Binding of soluble β-glucan and iC3b to the CR3 receptor induces the activation of the receptor and results in CR3-dependent cellular cytotoxicity (DCC) mediated lysis of iC3b-coated tumor cells [36,59]. Additionally, particulate β-glucans have also been demonstrated to induce CR3-mediated DCC in granulocytes. 

Yeast-derived particulate β-glucan has also been shown to activate dendritic cells and macrophages through a different pathway that involves Dectin-1, emphasizing the different mechanism of actions of different preparations of β-glucan [58]. The trafficking of oral β-glucan by intestinal macrophages to the spleen and lymph nodes, as described above, has been reported to activate DCs resulting in the activation and expansion of tumor-specific T cells that migrate to the tumors. Increased infiltration of activated DCs along with T cells and Th-1 cytokines in the tumors resulted in a significant reduction in the tumor burden [36]. Therefore, particulate β-glucan-activated DCs further enhance the anti-tumor efficacy by promoting inflammatory responses through the activation of both helper T cells and cytotoxic T cells. A bacterial β-glucan, Curdlan, has also been demonstrated to reprogram tumor-infiltrating DCs to induce Th1 T cell generation. Dectin-1 mediated DC reprogramming was shown to induce the generation of mucosal CD8 T cells expressing CD103 that accumulated in the tumors, significantly increased tumor necrosis, and thus inhibited tumor progression in humanized mouse models of breast cancer [60]. Curdlan has also been demonstrated to convert regulatory T cells (Tregs) into TH17 effector T cells both in vitro and in vivo using 4T1 mouse mammary tumor models [61] (Figure 1a). 

Macrophages are the major subsets of immune cells that infiltrate the TME and are known to play an important role in promoting cancer [62]. Pro-inflammatory macrophages or classically activated macrophages (M1) that infiltrate the TME during early stages of tumor growth have been known to switch to an anti-inflammatory or alternatively activated phenotype (M2) with tumor progression. These tumor-associated macrophages (TAM) have been correlated with poor prognosis [63] and poor survival [64]. Modulation of the TME through the switching of a TAM phenotype to a pro-inflammatory phenotype thus serves as a potential immunotherapy for cancer. Particulate β-glucan has been reported to bind to a C-type lectin receptor Dectin-1 expressed on TAMs, inducing a metabolic reprogramming, and thus conversion to a classically activated macrophage (M1) phenotype with enhanced antigen-presentation ability and Th1 cytokine production, such as IFN-γ [65] (Figure 1b). Another important subset of immune cells that plays a role in tumor progression are myeloid-derived suppressor cells (MDSCs). Administration of yeast-derived whole β-glucan particles (WGP) have been shown to reduce the polymorphonuclear-MDSC (PMN-MDSC) by inducing a respiratory burst and apoptosis of these immune sub-populations in sub-cutaneous Lewis lung carcinoma (LLC) tumors. Additionally, soluble β-glucan was also found to modulate the monocytic MDSC (M-MDSC) population by differentiating these subsets into potent antigen-presenting cells (APCs) that induced antigen-specific T cell responses in a dectin-1 dependent manner [66] emphasizing the ability of β-glucan to bridge the innate and adaptive immune systems (Figure 1c).

B-lymphocytes are another major subset of the adaptive immune system that have been reported to be influenced by β-glucan. Stimulation with both particulate and soluble β-glucan has been reported to activate B cells to induce the production of proinflammatory cytokines TNFα, IL-6 and IL-8 through dectin-1 mediated activation of transcription factors NF-κB and AP-1. Dectin-1-mediated proinflammatory cytokine secretion was further shown to stimulate neutrophil chemotaxis. The proliferation of B cells and antibody secretion, however, were not altered [67]. Another study investigating the effect of a soluble β-glucan on Dectin-1^+^CR3^−^ Human B-lymphoma cells observed an up-regulation of CD86 [68]. Therefore, these studies suggest an important role of Dectin-1 in the activation of B cells when stimulated with β-glucan.

## 5. β-Glucan and Metabolic Reprogramming

Understanding the metabolism of immune cells, stromal cells and tumor cells within the TME is of penultimate importance to understanding how to treat tumors effectively. For this reason, tumor metabolism has been a major focus of cancer research, where novel discoveries have been made that add significantly to the foundational cancer metabolism work done by Warburg in 1956 [69]. While, historically, the metabolism of the tumor cells themselves was the main focus of metabolic studies, it is now the case that focus is shifting towards the metabolism of the immune cells within the TME. Macrophage polarization, which plays a crucial role in the function of immune cells in the TME, is one such example, where M1 and M2 macrophages have been shown to have distinct metabolic signatures regarding the metabolism of folate, iron, glucose and phospholipids [70,71,72,73,74]. M1 macrophages are phenotypically characterized by glycolytic metabolism alone, whereas IL-4 induced M2 macrophages to show increased rates of oxidative phosphorylation. Regarding fungal β-glucan, when human monocytes are exposed to β-glucan, cells show proportionally increased glycolysis and decreased oxidative phosphorylation—a metabolic switch indicative of repolarization to the M1 phenotype [75]. Overall, the differential activation of coordinated metabolic and transcriptional responses is necessary for macrophage polarization, which ultimately has a profound impact on defences against tumor growth. Accordingly, fungal β-glucan has become an important part of the discussion regarding the metabolic reprogramming of myeloid cells, but especially regarding the concept of trained immunity.

Trained immunity stems from the observation that following exposure to certain vaccinations or infections, cells of the innate immune system incite enhanced immune responses upon heterologous re-infection, due to extensive metabolic and epigenetic reprogramming [76,77]. These epigenetic changes result in cellular activation, enhanced cytokine production and sweeping changes in metabolism [78]. Though several studies have shown that exposure to certain immune stimuli, such as LPS, results in immune tolerance, β-glucan has been identified as a potent inducer of trained immunity and immune stimulation through activation of the pattern recognition receptor (PRR) Dectin-1 [79,80,81]. As described by Cheng et al., trained monocytes have increased glycolysis, lactate production, and NAD+/NADH ratio, which reflects a shift in the metabolism of these trained monocytes. This shift of increased aerobic glycolysis was shown to be dependent on activation of the mammalian target of rapamycin (mTOR), due to a dectin-1-AKT-HIF-1α (hypoxia-inducible factor-1α) pathway. They showed that the inhibition of Akt, mTOR, or HIF-1α inhibited the induction of trained immunity in monocytes [75]. Additionally, Bekkering et al. showed that the induction of trained immunity by β-glucan is dependent on the activation of the cholesterol synthesis pathway, but not on the synthesis of cholesterol itself. Further, they show that mevalonate is an essential mediator of trained immunity, where mevalonate activates IGF1-R and mTOR, which results in histone modifications in inflammatory pathways [82].

Interestingly, experiments have also shown that the induction of trained immunity by fungal β-glucan is dependent upon autophagy. As shown by Buffen et al., the inhibition of autophagy using both pharmacological and genetic inhibition resulted in the blockade of trained responses [83]. Autophagy itself has been shown to be a crucial component of monocyte-macrophage differentiation. Stimulatory signals are requisite for the differentiation of monocytes, and in the absence of stimulation, monocytes will undergo apoptosis. When monocytes are triggered to differentiate, autophagy is actually induced, where it is thought that autophagy induction serves as an important transition from monocyte apoptosis to differentiation, and ultimately protects monocytes from cell death [84,85,86]. Especially in the context of cancer, where the survival and differentiation of monocytes into M1 macrophages is of extreme importance to tumor control, the induction of autophagy plays a crucial role in this process. Considering the interconnection of trained immunity and autophagy, there certainly needs to be more work done to untangle this intricate connection between trained immunity, autophagy and anti-cancer responses.

In the scope of cancer, the use of fungal β-glucan to induce trained immunity and enhanced immune responses could be very beneficial in immunotolerant states, such as cancer. In fact, known inducers of trained immunity are currently in use for the treatment of cancers, though when the treatments began, it was unknown how exactly these agents were working. Muramyl tripeptide, a synthetic lipophilic glycopeptide, and known inducer of trained immunity, is one such example that has shown promise in treating osteosarcoma through monocyte activation. Treatment with muramyl tripeptide was shown to result in macrophage activation and induction of cytokine and chemokine secretion by these activated macrophages that were responsible for the recruitment and stimulation of other immune cells [87]. Together this resulted in anti-tumor effects. The Bacillus Calmette-Guerin (BCG) vaccine, another known inducer of trained immunity, is widely recognized as a standard method of treating intermediate and high-risk non-muscle invasive bladder cancer [88]. Interestingly, the immunological mechanism responsible for the success of the therapy has been largely uncharacterized. Though researchers have generally understood macrophages and activated lymphocytes to be the driving factor behind clinical responses, it is quite likely that metabolic reprogramming of monocytes resulting in trained immunity is responsible for the therapeutic success [89,90]. This can be partly assumed based on the work of Alexander et al., where investigators determined that the systemic pretreatment of mice with particulate β-glucan significantly diminished the growth of metastatic-like B16 melanoma in the lungs, although they concluded that trained immunity by definition was not involved in this process. Β-glucan treatment was also shown to result in a dense myeloid immune cell infiltrate. Finally, their initial studies showed that though tumor suppressed HIF-1α expression, β-glucan treatment, known to induce HIF-1α as discussed above, successfully abrogated this observed suppression [91]. A follow-up study by the same group showed that the antitumor efficacy of yeast-derived β-glucan in a murine model of metastatic lung cancer is independent of canonically described β-glucan-dependent antitumor pathways, and was independent of adaptive immunity [92]. The scientific community’s understanding of β-glucan induced trained immunity is still in its infancy; however, as Netea et al. hypothesize, the induction of trained immunity using β-glucan alone or in combination with immunotherapy will prove a novel and effective strategy for cancer treatment [93].

## 6. Transcriptional Changes Induced by β-Glucan

Recognition of β-glucan as a PAMP by receptors expressed by innate immune cells like macrophages and dendritic cells activates a cascade of signaling that then induces the activation of key transcription factors resulting in the induction of effective anti-tumor responses and modulation of the TME [94]. Stimulation of myeloid cells with β-glucan has been shown to directly recruit and activate SyK kinase through dectin-1 followed by recruitment of Card9 to form the Card9/Bcl10/Malt-1 complex that activates the IκB kinase complex resulting in the activation of key transcription factor NF-κB. Activation of NF-κB then is able to initiate a pro-inflammatory signaling cascade [65]. The role of NF-κB in macrophages during tumorigenesis has been well-studied. Inhibition of NF-κB by targeting the IKK2 in myeloid cells has been shown to significantly reduce colon tumor incidence and size in a colitis-associated mouse model of cancer [95]. Another study in a mouse model of hepatocellular carcinoma showed a similar reduction of tumor incidence by targeting the liver macrophages using IκB-super repressor [96]. These studies indicate an important role for NF-κB in tumor progression. Interestingly, another study by Connelly et al. showed that modulation of NF-κB in macrophages before the seeding of tumor cells could prevent the metastasis of tumor cells to the lungs, emphasizing a role for modulating NF-κB to inhibit tumor progression [97]. Activation of NF-κB by β-glucan through dectin-1 singlaing, and the conversion of cells with a TAM phenotype to a pro-inflammatory phenotype thus unequivocally highlights the immunomodulatory potential of β-glucan.

## 7. β-Glucan as A Delivery Vesicle for Cancer Therapeutics

Particulate yeast-derived glucans are hollow, porous 2−4 µm spheres with an outer shell that mediates macrophage phagocytosis through dectin-1 signaling [98]. Considering the vast anti-tumor mechanisms of β-glucan, and that the glucan particles are hollow, they have become the subject of research aiming to utilize these properties to employ β-glucan as a vesicle to encapsulate, transport, deliver and release therapeutics within the TME, and specifically to macrophages. Several groups have already shown that it is possible to load β-glucan with different electrostatically-bound compounds, such as siRNA, DNA, protein, and small molecules in order to deliver these payloads to macrophages and DCs.

Targeting macrophages within the TME is particularly challenging for numerous reasons. First, within a tumor many of the macrophages are matured macrophages, which are end-stage cells that do not divide. As a result, using siRNA delivery systems that require the integration into dividing cells often is ineffective at targeting macrophages. For this reason, it becomes important to use a delivery system that involves that active uptake of the delivery vesicle. Second, macrophages being phagocytic cells are programmed to enzymatically degrade cellular components after phagocytosis. This can lead to the degradation of nucleic acids upon delivery, which ultimately inhibits the efficacy of therapeutically delivered siRNA. Using β-glucan to encase siRNA is one such way to effectively deliver siRNA intracellularly while maintaining the integrity of the nucleic acid. Macrophages are also not transient cells within a tumor, but can stay active within the TME where they impact adaptive immune responses for prolonged periods of time [99]. Accordingly, therapeutics meant to modify macrophage function must not be transient and also require a system that enables prolonged delivery of siRNA [100,101,102,103]. Thus, the delivery system must have minimal toxicity considering persistent administration may be necessary. For all of these reasons, using β-glucan to deliver siRNA is particularly attractive. Because tumors are often associated with the infiltration of macrophages into the TME, this can be exploited by loading small molecule drugs into macrophages for delivery to the TME [104]. Finally, it has been shown that oral delivery of these particles is not only safe, but effective.

In one study, researchers developed a glucan-based siRNA carrier system which was shown to assemble siRNA into uniformly distributed nanoparticles effectively. On administration, these β-glucan containing siRNA nanoparticles were shown to reduce gene expression of migration inhibitory factor (MIF), a cytokine and chemokine known to be upregulated in the TME in primary macrophages [105,106]. Upon IV injection, these nanoparticles were able to mediate a sustained reduction of MIF in tumor associated macrophages (TAMs) in a mouse model of 4T1 mammary cancer [103].

Another study, by Aouadi et al., characterized the use of β1,3-D-glucan encapsulated siRNA particles (GeRPs) which can be used an effective and specifically targeted oral delivery vehicle of siRNA directed against TNFα [107]. Though TNFα is associated with anti-tumoral responses, it is conceivable that other anti-inflammatory cytokines or pro-tumor signaling molecules could be targeted in a similar way in the TME [108]. The same group has also published literature regarding the loading of insoluble preformed nanoparticles loaded with doxirubicin inside β-glucan particles, or electrostatically bound to the surface. Because macrophages are terminally differentiated and non-dividing, they are resistant to doxorubicin, which works by blocking topoisomerase II in actively dividing cells [109]. Thus it was hypothesized that upon migrating to a tumor, macrophages would be able to release a payload of a lethal dose of doxorubicin, resulting in targeted delivery to rapidly dividing tumor cells [104]. 

β-glucan nanoparticles are another hot topic of research, where β-glucan is prepared using trifluoroacetic acid, which results in the generation of low molecular weight nanoparticles. These nanoparticles can then be used as both an immune modulator, and as a carrier of drugs or small molecules. In one particular example, glucan nanoparticles were effectively loaded with ssDNA [110]. 

A final example of manipulating the trafficking properties of β-glucan for the delivery of anti-cancer therapeutics comes from a study by Wang et al., where they conjugated a peptide antigen for a well-known cancer biomarker, MUC1, to β-glucan in order to construct a prospective cancer vaccine [111]. After successfully conjugating a MUC1 tandem repeat sequence to β-glucan, researchers injected mice with subcutaneous β-glucan alone, MUC1 peptide alone, or the synthetic conjugate. Following four immunizations, sera were collected on day 28 and analyzed for anti-MUC1 IgG antibodies (Abs). As assessed by ELISA, only the conjugate elicited high tiers of anti-MUC1 IgG Abs, indicating that the conjugation of MUC1 peptide to β-glucan was necessary for the stimulation of immune responses. Furthermore, only the sera of mice vaccinated with the construct exhibited positive reactivity to MCF-7 cells, a human breast tumor cell that highly expresses MUC-1. These results highlight the poor immunogenicity of the MUC-1 peptide alone and the marked efficacy of the peptide when conjugated to β-glucan. Together this suggests that by combining β-glucan with tumor antigens, we may be able to stimulate components of both the innate and the adaptive immune responses to result in more robust immune responses [112]. It also may suggest that β-glucan can be used as both a stabilizer and a carrier of peptides, which can ensure delivery to the lymph node and the generation of both humoral and memory responses [113]. (See Table 1 for a summary of studies using β-glucan as a delivery vehicle for cancer therapeutics)

## 8. β-Glucan in Combination Therapy

Immunotherapy has revolutionized the treatment of cancer with advances in clinical treatments, such as check-point inhibitors, adoptive immune cell transfer therapies, and chimeric antigen receptor T-cell therapies [114,115,116]. However, clinical efficacy is still not remarkable, with only a limited number of patients seeing a complete response to treatment. One major reason for the limited clinical success is the immunosuppressive nature of the TME [117]. The immunosuppressive TME is achieved by the recruitment of immunosuppressive cells, including macrophages, DCs, Tregs, and MDSCs, which render these tumors resistant to anti-tumor therapies. Modulation of the TME towards an immune-reactive state in combination with other anti-tumor therapies thus has potential in augmenting the anti-tumor immune response.

An extremely impactful aspect of β-glucan treatment is that it has been demonstrated to modulate the TME. Additionally, β-glucan is safe to use, non-immunogenic, and thus incites no other non-specific responses. It is well-tolerated even with a dose up to 10 mg/kg, induces specific responses through specific receptors and contains many side groups that allow further modification. These features establish β-glucan as a potent adjuvant [55]. Use of adjuvants not only with prophylactic vaccines, but also in therapeutic formulations or alone for immunomodulatory functions have been gaining popularity recently. Fungal β-glucan has been used as an adjuvant, along with other formulations, in both cancer prophylaxis and therapy, and as a single agent for immunomodulatory purposes, and are therefore referred to as immunoadjuvants [118].

β-glucan used in combination with monoclonal antibodies (mAb) has demonstrated efficient antitumor responses. A study by Hong et al., in 2003, assessed the efficiency of the combination of β-glucan with mAbs against naturally occurring tumor antigens G_D2_ ganglioside or recombinant human MUC1 and demonstrated a CR3-mediated granulocyte-dependent significant tumor regression with the combination of β-glucan and mAbs in five different mouse models of tumor [119]. Another study utilizing a combination of microparticle β-glucan and ovalbumin showed a dendritic cell-dependent T cell activation through increased upregulation of co-stimulatory molecules indicating enhanced antigen presentation by dendritic cells [112]. Also, a phase I clinical trial using a bivalent ganglioside vaccine in combination with β-glucan in patients with high-risk neuroblastoma demonstrated promising results. Twelve out of thirteen patients receiving the complete treatment were found to be relapse-free for a median of 32 months with significant antibody responses to the ganglioside vaccine.

Additionally, a disappearance of minimal residual disease (MRD) in 6 out of 10 patients was also observed [120]. Another phase I clinical trial for Metastatic Neuroblastoma employing β-glucan in combination with a monoclonal antibody, 3F8, has also reported the treatment to be well-tolerated by the patients along with antineoplastic activity [121]. Combination of β-glucan with a lung cancer vaccine 1650-G and GM-CSF as an adjuvant for a Stage I-IIA Non-Small Cell Lung Cancer (NSCLC) patients of phase I/II clinical trial also showed promising outcomes. The combination of β-glucan with the vaccine resulted in a robust immunologic response that was comparable to a DC vaccine indicating the combination treatment as a cheaper and promising alternative to DC vaccines for NSCLC [122].

A clinical trial combining Pembrolizumab with Imprime PGG for chemotherapy-resistant metastatic triple-negative breast cancer has recently been reported showing promising outcomes with a 64.2% patient survival for up to 12 months [123]. Another clinical trial with the same combination has also been reported to be ongoing for metastatic Non-small cell lung cancer (NSCLC) (NCT03003468). Treatment of M2 macrophages or DCs with soluble Imprime PGG β-glucan induces significant expression of PD-L1 (a marker for adaptive immune resistance) and CD86 (a co-stimulatory marker for T cell activation) [124]. Combination of Imprime PGG with anti-PD1 antibody (nivolumab) thus indicates a potential therapy to augment T cell activation. A study by Qiu et al., in 2016, observed increased T cell expansion and significantly reduced median tumor volume with the administration of Imprime PGG along with anti-PD1 mAB in a syngeneic CT-26 mouse model of cancer [125]. Considering that there still remains a significant population of patients that do not respond to checkpoint inhibitor therapies, there is a need for therapeutics that can influence the TME to be responsive. β-glucan has great potential in this regard as the upregulation of PD-L1 by β-glucan could cause tumor cells and macrophages to respond better to anti PD-L1 therapeutics, and the upregulation of CD86 indicates that despite the increased PD-L1 expression, these cells are still activated. 

Combination of fungal β-glucan and other chemotherapeutic agents have also been assessed in pre-clinical models, where they show significant anti-tumor responses when compared to chemotherapy treatment alone. Treatment of patients with advanced malignancies receiving chemotherapy with β-glucan in a phase I/II trial showed a beneficial effect of increased hematopoiesis, especially in patients with chronic lymphocytic leukemia (CLL) and lymphoma [126]. A phase II trial assessing the safety and efficacy of the combination of Imprime PGG with Cetuximab in patients with stage IV KRAS-mutant colorectal cancer demonstrated mild reactions of fatigue and headache to Imprime PGG and other observed toxicity related to cetuximab [127]. Another phase II trial for advanced non-small cell lung cancer patients assessed the safety and efficacy of β-glucan in combination with cetuximab/carboplatin/paclitaxel and demonstrated significantly enhanced objective response rate (ORR) as a first-line treatment [128]. Combination of Imprime PGG with rituximab in a phase II clinical trial for Relapsed Indolent Non-Hodgkin Lymphoma also showed an impressive response rate along with an increased release of pro-inflammatory cytokines related to an M1-macrophage phenotype, expansion and activation of tumor-infiltrating T cells [129]. These combinatorial strategies can be found summarized in Table 2. 

Together, these studies indicate an increasing interest in using β-glucan in combination with different anti-tumor therapies. Combination of β-glucan though, in its preliminary phase, seems to have promising outcomes emphasizing the need for more research in the future.

## 9. Conclusions

Though fungal β-glucan’s have been studied and used as an immune-stimulant for centuries, there are many aspects of β-glucan biology that are only now being brought to light which are profoundly shaping immunological paradigms. Specifically, the ability of β-glucan to incite the metabolic reprograming and epigenetic regulation of myeloid cells and macrophages is a revelation in the fields of both infectious disease and cancer. As the concept of trained immunity in the context of metabolic reprograming is relatively novel and unstudied, we expect there to be a great deal of interest going forward in how innate memory responses may play a role in cancer immunoprevention and treatment. Additionally, though it has long been a goal of medicine to develop tumor vaccines and many exist in clinical trials, to date, vaccination against HPV remains the sole example of the widespread and successful use of a tumor vaccine. It also remains the case that preventative strategies for thwarting cancer development are few and far between. A growing body of literature is indicating that the use of β-glucan as an adjuvant has the potential to shape the development of anti-tumor vaccines. Especially considering its proven safety and low toxicity, it is very likely that research coming out now is only the beginning of the various ways that β-glucan will be used to augment immune responses and promote adaptive memory immunity against tumors. Finally, the combinatorial use of β-glucan with immunotherapy and chemotherapy is beginning to show great promise in improving patient morbidity and mortality. More research and clinical trials are certainly needed in order to further understand the beneficial effects of β-glucan in combination with the mainstay drugs used in cancer treatment/prevention. Especially given the newly embraced functionality of β-glucan as a vesicle to transport other anti-cancer therapeutics, it appears that β-glucan’s anti-cancer properties are truly multifaceted. Finally, given the numerous and diverse sources of β-glucan along with the various ways of processing and purifying it, one of the greatest challenges in β-glucan-related research remains the standardization and proper characterization of the molecules themselves.

## Figures and Tables

**Figure 1 ijms-20-03618-f001:**
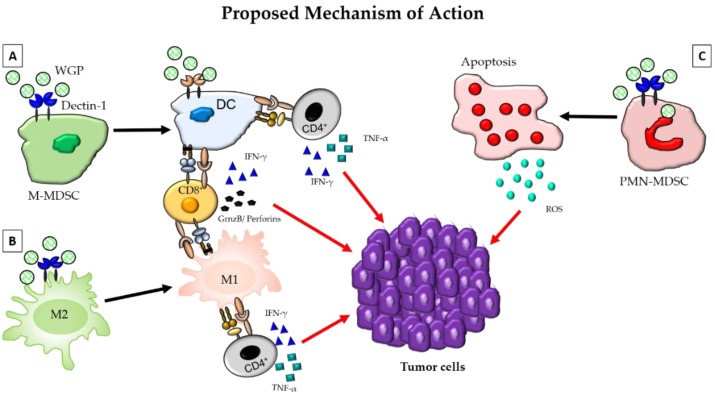
The modulation of the immune cells in the tumor microenvironment by β-glucan. β-glucan binds to the Dectin-1 receptors expressed on cells of the myeloid lineage, and will then be phagocytosed. In 1a, β-glucan can be seen binding to Dectin-1 on an M-MDSC. Binding to the M-MDSC will cause the M-MDSC to switch from a suppressive phenotype, to a DC phenotype that can act as an APC. This dendritic cell (DC) will then activate CD4+ and CD8+ T-cells, where CD4+ T-cells will secrete pro-inflammatory cytokines, such as TNFα and IFN-γ, and CD8+ T-cells will secrete Granzyme B, perforins and IFN-γ. The secretion of these pro-inflammatory cytokines by CD4+ and CD8+ T-cells will lead to the destruction of tumor cells. Similar to 1a, in 1b, β-glucan induces the polarization of suppressive M2 macrophages into inflammatory M1 macrophages. M1 macrophages will then activate Th1 type T-cells, leading to damage to the tumor cells through the secretion of pro-inflammatory cytokines by CD4+ and CD8+ T-cells. Finally, in 1c, β-glucan will bind to the Dectin-1 receptor on polymorphonuclear (PMN)-MDSCs and cause apoptosis of the cell. As the cell undergoes apoptosis, it will produce ROS that will ultimately target the tumor cells, leading to tumor cell death. Overall, these mechanisms together convert a suppressive tumor microenvironment (TME), to an inflammatory TME that has a greater potential to induce killing of tumors.

**Table 1 ijms-20-03618-t001:** The use of β-glucan to deliver cancer therapeutics.

Therapeutic Delivered	Glucan Utilized	Delivery Mechanism/Mechanism of Action	Results of Study	Reference
Anti-MIF siRNA	β-(1,3)(1,4)-D-glucan	A glucan based siRNA carrier system (BG34-10-Re-I) delivered IV that can effectively assemble siRNA into uniformly distributed nanoparticles that are delivered to macrophages.	Injection of the nanoparticles into Balb/c mice bearing 4T1 mammary tumors resulted in MIF reduction in TAMs	[106]
Anti-TNFα siRNA	β-(1,3)-D-glucan	β1,3-d-glucan-encapsulated siRNA particles (GeRPs) were used as oral delivery vehicles. After oral consumption, particles were phagocytosed by macrophages and DCs in the GALT, then trafficked to other reticuloendothelial system tissues.	Oral gavage of mice with GeRPs containing as little as 20 µg kg^−1^ siRNA directed against TNFα depleted its messenger RNA in macrophages recovered from the peritoneum, spleen, liver and lung, and lowered serum TNFα levels.	[107]
Doxorubicin	β-(1,3)-D-glucan	Glucan particle nanoparticle formulations were made with fluorescent anionic polystyrene nanoparticles. Mesoporous silica nanoparticles (MSNs) containing doxorubicin were bound to the glucan nanoparticles	Doxorubicin loaded glucan particles effectively delivered doxorubicin into phagocytic ells resulting in enhanced cell-cycle arrest in vitro.	[104]
ssDNA	Zymosan	β-glucan nanoparticles (GluNPs) were prepared by slicing β -glucan into low molecular weight using various concentrations of Trifluoroacetic acid (TFA). ssDNA was then inserted into the glucan triple helix structure.	The group successfully produced a low molecular weight β-glucan containing ssDNA. The particles have not yet been used to test therapeutic efficacy.	[111]
MUC-1 peptide antigen	β-(1,3)-D-glucan	A Muc-1 peptide antigen was conjugated to β-glucan with the goal of creating a cancer vaccine candidate. It was hypothesized that recognition, and uptake of the conjugate by myeloid cells may activate innate immunity while the presentation of the MUC1 epitope may activate adaptive immunity, leading to overall more robust responses to tumors displaying high levels of MUC-1	Mice subcutaneously receiving the conjugate showed high anti-MUC1 antibody titers cross-reactive with the MCF-7 tumor cell line, as well as high levels of IL-6 and IFN-γ in sera. Overall the conjugate was able to elicit potent immune responses and immunogenicity of the MUC-1 peptide enhanced through conjugation to β-glucan.	[113]

**Table 2 ijms-20-03618-t002:** Ongoing clinical trials using β-glucan in combination therapy for cancer treatment.

Combination	Type of Cancer	Status	Clinical Trial ID	Outcomes
β-Glucan with OPT-821 and vaccine therapy(β-glucan and OPT-821 as immunoadjuvants to modulate the immune system and vaccine to produce an effective immune response against the tumor cells)	Neuroblastoma	Phase I/II	NCT00911560	No dose-limiting toxicity reported at 150 µg/m^2^ of OPT-821, relapse-free survival was reported to be 80% ± 10% at 24 months, significant antibody response against GD2 and/or GD3 (OPT-821) in 12 out of 15 patients [120]
Pembrolizumab with Imprime PGG (Imprime PGG enhances the sensitivity to checkpoint inhibitors by activating the innate and adaptive immune responses)	Chemotherapy-resistant metastatic triple-negative breast cancer	Phase II	NCT02981303	15.9% Overall response rate (ORR); 25% disease control rates for >24 weeks; 13.7 months overall survival with 79% patient survival up to six months, 71.5% survival up to nine months and 64.2% patient survival up to 12 months [123].
Pembrolizumab with Imprime PGG	Metastatic Non-small cell lung cancer	Phase I/II	NCT03003468	Trial ongoing
Monoclonal antibody 3F8 with β–glucan(3F8 blocks tumor growth and β-glucan stimulates the immune system to kill tumor cells)	Metastatic Neuroblastoma	Phase I	NCT00492167	Combination therapy of 3F8 along with β-glucan has been reported to be well-tolerated with antineoplastic activity [121].
Rituximab and β-glucan(rituximab locates cancer cells enabling cancer cell killing or delivering cancer-killing substances to the cancer site and combination with β-glucan increases the effectiveness of rituximab by making cancer cells more sensitive to the antibody)	Relapsed or Progressive lymphoma or Leukemia, or Transplantation-related Lymphoproliferative Disorder	Phase I	NCT00087009	Trial Ongoing
Rituximab and Imprime PGG(Imprime PGG activates innate immune cells through CR3 and combination with rituximab improves the responses through complement-dependent cytoxicity)	Relapsed Indolent Non-Hodgkin Lymphoma	Phase II	NCT02086175	Combination of β-glucan with rituximab was well-tolerated with 46% observed response rate, resulted in pro-inflammatory cytokines related to M1-macrophage phenotype, increased antigen presentation and expansion and activation of tumor-infiltrating T cells [129].
Lung cancer vaccine 1650-G and GM-CSF with β–glucan(1650-G is a vaccine comprising killed allogeneic tumor cells that act as tumor antigens, GM-CSF acts as an adjuvant)	Stage I-IIA Non-Small Cell Lung cancer (NSCLC)	Phase I/II	NCT01829373	Combination treatment is reported to be safe with a robust immunologic response in 6 out of 11 patients, and the kinetics of the responses were observed similar to that with DC vaccines indicating a promising and cheaper multivalent vaccine for NSCLC [122].
Cetuximab/Carboplatin/Paclitaxel in combination with Imprime PGG	Advanced Non-Small Cell Lung Cancer	Phase II	NCT00874848	The combination treatment was well-tolerated and showed improved overall response rate (ORR) [128].

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
