# Peer review of "Yeast-Derived β-Glucan in Cancer: Novel Uses of a Traditional Therapeutic"

_ijms, 2019, doi:10.3390/ijms20153618_

Reviewer 1 Report

There are several minor points for revision:

Title: I recommend to use "Fungal β-glucans" instead "β-glucan" in many cases including the title because this is a group of structurally different polysaccharides and   there are cereal β-glucans as well. Also, it is not good to use the same word twice in the title. So, corrected title could be "Fungal β-glucans against cancer: Novel uses of a traditional treatment."

Keywords: use "fungal β-glucans" instead "β-glucan"

Page 2, lines 64-81: Structural diversity of fungal β-glucan is much higher than that described in this part. It should be expanded, see for example Synytsya A, Novák M. Structural diversity of fungal glucans. Carbohydrate Polymers 2013, 92(1):792-809.

Author Response

Thank you so much for taking the time to review our paper. We appreciate your time and your comments. Everything that was changed has been made red in the text of the paper.

We have changed the title to read " Yeast-derived beta-glucan in Cancer: Novel Uses of a Traditional Therapeutic"

The key words have been changed to fungal beta glucans

The word "fungal" has been placed before beta-glucan in several instances (all highlighted in red) in order to provide further clarity that the focus is on fungal beta glucan. Lines: 99, 195, 198, 277, 282, 302, 315, 457, 495, and 517

Further, a sentence was added in the introduction indicating that the focus will be fungal beta glucan (line 25-26)

The diversity of fungal beta glucan was expanded upon as requested. These changes can be seen in lines 66-84.

Reviewer 2 Report

Geller et.al in this review have reported the anticancer and oncoimmunotherapeutic role of the natural product beta-glucan. Though the review is very informative and has merit, I recommend the following changes which can potentially improve it before publication.
1) please provide a scheme showing the mechanisms of beta-glucan on immune cells modulation in the tumor microenvironment.

2) please provide a table showing combination therapies of beta-glucan with other anticancer/oncoimmunotherapeutic agents

3) please provide a table showing beta-glucan as delivery vehicles for cancer therapeutics

4) please provide a table for Clinical trial results (if needed could be merged with the table mentioned in 3).

Author Response

Thank you so much for taking the time to review our submitted paper. We deeply appreciate it. A written response is provided below each suggested change from the reviewer. All changes to the text itself have been made in red.

1) please provide a scheme showing the mechanisms of beta-glucan on immune cells modulation in the tumor microenvironment.

A scheme has been added. It is listed as Figure 1 and can be found on page 6.

2) please provide a table showing combination therapies of beta-glucan with other anticancer/oncoimmunotherapeutic agents

We made a table that combined 2+4, where we listed combination therapies using beta-glucan with other anti-cancer agents, and also listed the outcome and status of the trial. It can be found on page 12-13 and is listed as Table 2: Ongoing clinical trials using b-glucan in combination therapy for cancer treatment

We felt it was best to combine combination therapies being used and clinical results or current status of that trial for clarity. Most therapies are currently in an early stage, so we also provided the clinical trial identifier number so readers may follow up on where the trial stands now.

We also found a few very recent studies that we had missed and included these both in the paper and in the table.

3) please provide a table showing beta-glucan as delivery vehicles for cancer therapeutics

This is a very new technology and thus has not been tested in humans yet, so there are no clinical trial results yet. For this reason we left this as its own table. It can be found on pages 9+10, and is listed as:

Table 1: The use of β-glucan to deliver cancer therapeutics.

4) please provide a table for Clinical trial results (if needed could be merged with the table mentioned in 3).

This was included with number 2 and can be found in table 2 on page 12-13.